# Variable *Legionella* Response to Building Occupancy Patterns and Precautionary Flushing

**DOI:** 10.3390/microorganisms10030555

**Published:** 2022-03-03

**Authors:** William J. Rhoads, Meril Sindelar, Céline Margot, Nadine Graf, Frederik Hammes

**Affiliations:** 1Department of Environmental Microbiology, Eawag—Swiss Federal Institute of Aquatic Science and Technology, 8306 Deubendorf, Switzerland; merils@student.ethz.ch (M.S.); celine.margot@eawag.ch (C.M.); nadine.graf@eawag.ch (N.G.); 2Department of Biology, ETH Zurich, 8092 Zurich, Switzerland; 3Department of Environmental Systems Science, ETH Zurich, 8092 Zurich, Switzerland

**Keywords:** *Legionella*, water demand, water age, flushing, stagnation, COVID-19, lockdown, boiler, recommissioning, water temperature

## Abstract

When stay-at-home orders were issued to slow the spread of COVID-19, building occupancy (and water demand) was drastically decreased in many buildings. There was concern that widespread low water demand may cause unprecedented *Legionella* occurrence and Legionnaires’ disease incidence. In lieu of evidenced-based guidance, many people flushed their water systems as a preventative measure, using highly variable practices. Here, we present field-scale research from a building before, during, and after periods of low occupancy, and controlled stagnation experiments. We document no change, a > 4-log increase, and a > 1.5-log decrease of *L. pneumophila* during 3- to 7-week periods of low water demand. *L. pneumophila* increased by > 1-log after precautionary flushing prior to reoccupancy, which was repeated in controlled boiler flushing experiments. These results demonstrate that the impact of low water demand (colloquially called stagnation) is not as straight forward as is generally assumed, and that some flushing practices have potential unintended consequences. In particular, stagnation must be considered in context with other *Legionella* growth factors like temperature and flow profiles. Boiler flushing practices that dramatically increase the flow rate and rapidly deplete boiler temperature may mobilize *Legionella* present in biofilms and sediment.

## 1. Introduction

Foundational support for the widely accepted belief that building water stagnation facilitates *Legionella* colonization and growth is far less convincing than assumed of such a central dogma. For instance, the most widely referenced study in support of this belief reported *Legionella* numbers in just two room temperature boilers that were unused for 18 months and actually did not report consistent evidence of *Legionella* growth [1]. The most referenced contrarian study reported less *Legionella* in stagnant pipes compared to a continuously recirculating loop, but the flowing condition had very little water exchange (5%), and high *Legionella* numbers were still reported in the stagnant condition [2]. Moreover, the specific conditions examined in these studies have little to do with typical hot- or cold-water installations in buildings and ignore the role of other critical operational parameters. 

As an example, the interaction between water temperature and stagnation can cause different trajectories in *Legionella* growth. In a controlled pilot-scale experiment, stagnation contributed to growth when it provided relief from inhibitory conditions (e.g., high temperature dissipating) but also resulted in no or limited growth when nutrients were scarce (too infrequent water exchange); similarly, flow limited or eliminated growth when it delivered inhibitory conditions (hot water, >48 °C in that study) but contributed to growth when inhibitory conditions were absent (water not hot enough) [3,4]. This variability is born out in many field studies, with suboptimal hot water recirculation temperature [5], inadequate delivery of disinfectants [6], and intermittent flow to distal locations with long pipe runs [7] coinciding with high *Legionella* positivity and numbers. 

The apparent contradictions in the impact of stagnation are complicated by the all-encompassing nature in which the term is typically used [8]. “Stagnation” has been used to describe a range of plumbing conditions including no water flow or exchange [1], existence of dead-end pipes [9], intermittent use at individual outlets [7], and generally elevated water residence time caused by low water demand and/or over-sized components [4]. It has also been used to refer to durations lasting one night to more than a year. Even in the study that demonstrated stagnant conditions supported fewer *Legionella* than flowing conditions referenced above [2], the limited water exchange in the system could, by one definition, classify the entire system as stagnant and diminish the conclusion that stagnation does not support growth. Thus, we reserve the term stagnation to refer only to the absence of water demand and flow (such as in dead-legs), and use other descriptive terms (e.g., reduced or low water demand, recirculating water with no exchange) to describe the nature of water demand and flow patterns. 

Though periods of low water demand in buildings are not uncommon during holidays or due to seasonal use, concerns over widespread reductions in water demand during “stay-at-home” orders to prevent the spread of the 2019 novel coronavirus disease (COVID-19) motivated academics [10], the health community [11], water regulators [12], practitioners [13], and news media [14] to warn building managers and consumers of the potential negative impact of decreased water demand on *Legionella* numbers in building plumbing systems. It was hypothesized that if the colloquial “COVID stagnation” did cause a significant increase in *Legionella* or other opportunistic pathogens (OPs), it may lead to an epidemic of additional disease considering that bacterial co-infections caused the most fatalities in past influenza pandemics (in 1918, 1957, 1968, and 2009) [15,16]. In the absence of evidence-based preventative measures, many practitioners adopted simplified guidance to flush their water systems as a precautionary measure. However, existing flushing recommendations generally do not account for the water quality delivered during flushing, are not evidence-based, and do not consider possible unintended consequences. In addition, because occurrence of OPs goes virtually unassessed in non-healthcare buildings, even though OPs are frequently detected in water systems [17], retrospectively evaluating the success of such flushing measures is difficult.

Here, we present illustrative field-scale research on the effect of three- to seven-week periods of reduced water demand associated with low building occupancy on *L. pneumophila* occurrence. Through regular measurements and controlled experiments within a building that has historically been colonized by *Legionella*, we document varied response of *Legionella* growth to reductions in water demand during the lockdown associated with the first wave of COVID-19 in Switzerland and winter holiday breaks. Importantly, this work represents one of the very few extensively documented non-healthcare buildings before, during, and after COVID stagnation, providing more context by documenting changes in *L. pneumophila* numbers and nuancing the foregone conclusion that stagnation causes *Legionella* growth. We also evaluate flushing practices implemented after extended periods of low water demand and document potential unintended consequences of extensive flushing practices in some instances. 

## 2. Materials and Methods

### 2.1. Site Description

The 8-story Swiss Federal Aquatic Research Institute (eawag) laboratory and office buildings have approximately 150 outlets supplied with non-chlorinated municipal drinking water by the Water Supply Duebendorf. Water is heated in a central 1000 L boiler and then either pumped in a recirculation loop and distributed to laboratory building floors via passive recirculation loops or unidirectionally supplied to lavatories and other public spaces (kitchenettes, water fountains) in the laboratory and office buildings by risers (Figure 1). During this study, the boiler was operated at 45 °C from 0:00–23:59 five days per week and at 60 °C on Tuesdays and Thursdays. At night (19:00–05:00), the recirculation pump was turned off, but the boiler maintained its target setpoint. Outlets are connected to the floor loops (laboratory outlets) or hot water supply risers (public area outlets) with a 16 mm diameter steel pipe from the riser or floor loop to the room location, a 16 mm PEX-c pipe from the room entry to stopcocks, and 8 mm polymer tube with braided sheath that connects hot and cold stopcocks to outlets. (Figure 1; Appendix A). 

### 2.2. Low Water Demand Case Study and Experiments

The Eawag research buildings were found to be contaminated with *L. pneumophila* in 2016, which motivated a three-year investigation tracking the success of various interventions [18]. Throughout 2019, the last planned year of the investigation, *L. pneumophila* numbers were consistently low in first draw and flushed samples, with only two first draw samples violating the Swiss federal threshold of 1000 cfu/L (data not shown). Thus, our buildings provide a unique opportunity to monitor the impact of periods of reduced water demand. 

During routine monitoring as part of the case study, first draw (1 L) and 5-min flushed (~10 Lpm flow rate, 250 mL) samples were collected in autoclaved glass bottles throughout the laboratory and office buildings. The first 1 L represents water held in the outlet, stopcock connector tubes, service pipe from the floor loop or riser to room, and approximately 700 mL of the floor loop or riser; the flushed samples represent the main recirculating loop and boiler. A summary of all routine sampling sets is located in the Appendix A (Appendix A).

#### 2.2.1. COVID Lockdown

In February 2020, during routine data collection approximately 2 weeks before COVID Lockdown was initiated, first draw (*n* = 14) and 5-min flushed (*n* = 7) samples were collected. After 7 weeks of lockdown with building occupancy limited to essential personnel (5% occupancy), first draw (*n* = 40), and 5-min flushed (*n* = 19) samples were collected from outlets at the beginning, middle, and end of each floor loop, outlets in the laboratory building served by a riser, and outlets in the office building served by a riser.

#### 2.2.2. Winter Holiday Breaks

As there were no operational or temperature setting changes occurring in the system starting in January 2019, 3-week winter holiday breaks from December 2019 to January 2020 and December 2020 to January 2021 were evaluated as additional periods of reduced water demand. First draw and 5-min flushed samples collected during routine monitoring from July 2019 to January 2020 were subset to represent the period before (nFD = 65; n5 min = 37) and after (nFD = 18; n5 min = 7) the 2019 winter holiday. A more controlled sampling scheme was used to evaluate the 2020 winter holiday, where first draw (*n* = 34) and 5-min flushed samples (*n* = 12) were collected from the same outlets before and after the break. 

#### 2.2.3. Controlled Stagnation

Complete stagnation was imposed on 13 laboratory water outlets on both C- and E-floors beginning 6 July 2020 to simulate 1- to 4-week stagnation events. These floors were chosen because they had characteristically very poor and very good passive recirculation, respectively. One day before the stagnation experiment began (when the boiler was operated at 45 °C), each outlet was flushed with hot water for two minutes to establish similar initial conditions, sampled to represent time-zero, and randomly assigned a 1-, 2-, or 4-week stagnation period. After the assigned period, the outlet was opened, approximately 50 mL discarded (the volume held within the fixture), and the next 300 mL was collected in a sterile glass bottle to target water that resided in the pipe between the outlet and floor loop, where stagnation was occurring and where the majority of bacterial regrowth occurs in the laboratory building. 

### 2.3. Recommissioning Case Study and Experiments

#### 2.3.1. Recommissioning Flushing

After the 7-week COVID-19 lockdown, the plumbing was recommissioned according to a fact sheet issued by the Swiss Gas and Water Industry Association supported by the federal government prior to reoccupying Eawag [19]. This consisted of flushing each outlet until constant temperature was reached or for 2 min (whichever duration was shorter) on a day when the boiler was operated at 60 °C to maximize the potential for thermal disinfection. Afterwards, weekly samples were collected for 5 weeks (6 May to 8 June 2020). First draw and 5-min flushed samples were collected from the same outlets each week (*n* = 18 “regular” samples; nFD = 10, n5 min = 8) and outlets that were randomly chosen (*n* = 8 “random” samples; nFD = 6–8, n5 min = 0–4) to detect potential bias that flushing outlets weekly may have caused (Appendix A). Then, two additional samplings occurred (27 July and 30 October 2020; nFD = 6–12; n5 min = 2–6) to track numbers over a longer period. 

#### 2.3.2. Boiler Turnover Flushing

High water demand events that result in boiler volume turnover were simulated on two separate dates. On the first flushing date, one outlet was flushed at 5 Lpm into a sterile beaker and allowed to overflow to the drain for 400 min (~200% boiler volume turnover). The water flow was serially subsampled into 1 L bottles using a peristaltic pump, with each sample representing a composite of approximately 25% of the boiler volume. During the second boiler flushing, all outlets on the F-floor were opened to achieve a total of ~95 Lpm for 21 min (~200% boiler volume turnover). The flow was subsampled at one outlet (at ~0.38 Lpm), again with each 1 L composite sample representing approximately 25% of the boiler volume. *L. pneumophila* was cultured and *Legionella* spp. and *L. pneumophila* gene copy numbers were quantified using ddPCR in each sample. Before and weekly for 3–4 weeks after each high water demand event, 8 first draw and flushed samples were collected for *L. pneumophila* culture to assess the impact of the turnover event. 

### 2.4. Sample Processing and Analysis

#### 2.4.1. Flow cytometry and Legionella culture

Total cell counts (TCC), intact cell counts (ICC), and *L. pneumophila* (Lp) culturability were quantified in all samples, except the routine monitoring samples for which only *L. pneumophila* culture was performed. TCC and ICC were measured using a CytoFLEX (Beckman Coulter, Brea, CA, USA) flow cytometer in 250 µL aliquots stained using SYBR^®^ Green I (SG, Invitrogen AG, Basel, Switzerland; 10,000× diluted in Tris buffer, pH 8) for TCC or SYBR Green-Propidium Iodide (SGPI; SG with additional propidium iodide in a final concentration of 0.3 mM) for ICC. Stained cells were incubated for 10 min at 37 °C prior to analysis [20]. *L. pneumophila* was quantified using the Legiolert most probable number assay (IDEXX Laboratories, Inc., Westbrook, ME, USA) according to manufacturer protocols [IDEXX]. Legiolert is a liquid-based culture most probable number (MPN) method that correlates well with standard culture methods and has a low (3–4%) false positivity rate [21,22,23,24,25,26,27]. 

#### 2.4.2. Sample Filtration and DNA Extraction

Water sample aliquots from the boiler turnover flushing experiments (*n* = 21) samples were filter concentrated (0.2 μm polycarbonate membrane filters), fragmented using a flame-sterilized scalpel, DNA extracted, and *Legionella* spp. and *L. pneumophila* gene copies quantified using digital droplet polymerase chain reaction (ddPCR). 

DNA extraction was carried out using an adapted FastDNA Spin Kit (MPBiomedicals). Briefly, fragmented filters were combined with 6 µL Lysozyme (50 mg/µL) and 294 µL 1X TE buffer and incubated 1 h at 37 °C mixing at 300 rpm; afterwards, 30 µL Proteinase K (20 mg/mL) and 300 µL CLS-TC were added and incubation continued at 56 °C mixing at 300 rpm for an additional 30 min; FastDNA Spin kit beads and 600 µL chloroform (isoamylalcohol, 24:1 suitable for nucleic acid purification) were added, and samples were vortexed on a tube shaker at maximum speed for 5 min. Samples were then centrifuged at 14,000× *g* for 10 min, and approximately 750 µL of the upper aqueous solution was added to an equal volume of binding matrix; then, instructions from the FastDNA Spin kit were followed. A DNA extraction negative (unused filter processed identically to samples) and positive control (environmental source of culture-confirmed *L. pneumophila*) was included each time DNA extraction was performed (Appendix A). 

#### 2.4.3. Digital Droplet PCR (ddPCR)

*Legionella* spp. (ssrA) and *L. pneumophila* (mip) were measured using a digital droplet polymerase chain reaction (ddPCR) duplex assay. Gene target primers and probes were based on previously published assays validated to ISO SO TS12869:201 [28,29,30] and adapted for the ddPCR platform (Stilla, Villejuif, France). Briefly, each 25 µL reaction contained 1X PerfeCT a Multiplex ToughMix 5X (Quantabio), 0.6 µM of ssrA and 0.4 µM of mip gene forward and reverse primers, 0.15 µM of each probe, 100 nM Fluorescein (Sigma Aldrich), and 5 µL of DNA template. Primer and probe sequences, master mix composition, and thermocycling conditions can be found in the (Appendix A). A ddPCR reaction negative control (DNAse free water) was included for each batch of master mix prepared and was always negative. A ddPCR reaction positive control (Centre National de Référence des Légionelles) was included on each thermocycling run (Appendix A). Each run consisted of reactions loaded into three Sapphire 4-well chips (12 reactions per thermocycling run). Each batch of master mix consisted of three runs executed simultaneously (36 reactions per batch). Droplet formation and PCR thermocycling were performed using a Stilla geode and read using a Prism6 analyzer with Crystal Reader software imaging settings pre-set and optimized for PerfeCT multiplex master mix. Droplets were analyzed using Crystal Miner software. Only wells with a sufficient number of total and analyzable droplets, as well as a limited number of saturated signals, were accepted according to the Crystal Miner software quality control. Positive droplets were delineated using polygons, with positive wells being considered as those resulting in at least three droplets within the polygon. The limit of detection (LOD) was determined by the gene copy concentration that had a detectable signal in 90% of replicate wells, which was 12 gc/reaction (2.4 gc/µL template). The limit of quantification (LOQ) was determined by the gene copy concentration that had <25% residual standard deviation (RSD = Standard Deviation/Average × 100%) of replicate wells, which was 25 gc/reaction (5 gc/µL template) for both *Legionella* spp. and *L. pneumophila*. Any sample with significant rain was diluted 1:10 and rerun. Assay linearity and determination of the LOD and LOQ are presented in the Appendix A (Appendix A).

#### 2.4.4. Water and Pipe Surface Temperature 

Water temperature after sample collection and during flushing experiments was measured with a temperature datalogger (Testo Type K EF with 175T3 logger, Lenzkirch, DE, Germany). Pipe surface was measured with temperature loggers (Switrace i-Plug IPMT8-X3, Mendrisio, CH, Switzerland) attached to pipe segments underneath existing insultation at appropriate locations for a subset of monitoring periods to quantify temperature profiles in the main recirculating line, passive floor loops, and within individual laboratories (Appendix A). Boiler water temperatures were directly monitored with an in-line temperature probe installed at the top and bottom of the boiler throughout the study. 

### 2.5. Data Analysis

*L. pneumophila* numbers are reported as log(MPN or gene copy +1). Boxplots were generated using ggplot2 in R Studio (version 4.1.2). Parametric (*t*.test) or non-parametric (wilcox.test) tests were applied, as appropriate, to determine differences in measured concentrations between two groups; Kruskal-Wallis test with Bonferroni correction for multiple comparison (dunn.test) were used to determine differences between multiple categories. Chi-squared test (chisq.test) was used to determine the difference in proportion of positive samples between two groups. Multiple linear regression (lm) on log-transformed *L. pneumophila* culture data was used to identify the most influential regression coefficients. Significance was determined at *p* < 0.05.

## 3. Results

In the results below, we first establish the baseline *L. pneumophila* numbers in the building during normal occupation (and prior to the COVID-19 lockdown) followed by the impact of three- to seven-week periods of reduced building occupancy. We then show the impact of COVID-19 recommissioning flushing and the controlled follow-up boiler flushing experiments.

### 3.1. Low Baseline L. pneumophila Numbers during Normal Building Occupancy

In the period of normal building occupancy during 2019, when no major operational changes were implemented to the system, *Legionella* levels were low (median < 10 MPN/L; Figure 2A). Log-transformed multiple linear regressions indicated that the interaction between system (Floor Loop vs. Riser) and draw (first-draw vs. 5-min flushed; *p*-value = 0.0017), and outlet floor (B- to H-Floor, *p*-value < 0.004) were the most meaningful predictors of *L. pneumophila* during routine monitoring in 2019. For instance, there were no significant differences in *L. pneumophila* sample positivity rate (Chi-Squared, *p*-value = 0.31) or culture numbers (Wilcoxon, *p*-value = 0.090) between the passively recirculating floor loops and risers, or between first draw and flushed samples overall (C-S *p*-value = 0.54; M-W *p*-value = 0.22); however, *L. pneumophila* numbers were higher in first draw compared to flushed samples in outlets served by passive floor loops (*p*-value = 0.036), but not in outlets served by risers (*p* = 0.50). Interestingly, the floor with the highest overall positivity rate (E-floor, 83% positive, *n* = 12) had very good passive recirculation exchange with the main recirculating loop, while the floor with the lowest positivity rate (C-floor, 29% positive, *n* = 21) had the poorest exchange (Appendix A); E-floor also had 1.5-log higher median *L. pneumophila* than C-Floor (Dunn Test with Bonferroni correction, *p*-value = 0.011). This indicates some level of variation in the factors impacting *Legionella* growth in this system, and these data serve as a baseline to evaluate the impact of periods of low water demand.

### 3.2. Low Water Demand Case Study and Experiments

#### 3.2.1. No Legionella Increase Observed after 7-Week COVID-Lockdown

Two weeks after a routine monitoring sample set was collected in February 2020, Eawag implemented a work-from-home policy to support the effort to slow the spread of COVID-19. The lockdown lasted 7-weeks with only essential building occupancy (~5%). While *Legionella* positivity (100% in February 2020) numbers before lockdown were relatively high (Figure 2A,B), contrary to expectations, median *L. pneumophila* culture numbers decreased by 1.37 to 4.14-logs after lockdown (Figure 2B, Kruskal-Wallis, *p*-value < 0.001). Though the first draw samples tended to have a higher variation in levels, there were no statistically significant differences between first draw and flushed samples before or after lockdown (*p*-values 0.18–0.59).

#### 3.2.2. Variable Legionella Response to Winter Holiday Breaks

*Legionella* responded inconsistently to periods of low water demand that occurred during the winter holiday breaks prior to and after the COVID-19 lockdown. Before the winter break in 2019, *L. pneumophila* levels were low (median < 10 MPN/L, positivity 44%), but after the winter break in 2019, median culture numbers increased by >2.3 to 4.1-logs in first draw and flushed samples (Figure 2A; positivity 91%; Kruskal-Wallis, *p*-values < 0.001). In contrast, before the winter break 2020, no culturable *L. pneumophila* was detected and there was also not a significant increase in *L. pneumophila* culture numbers after the 3-week period of low water demand (Kruskal-Wallis, positivity 0% before and 8% after break; *p*-value = 0.14; Figure 2C). The four samples that were positive after the break in January 2021 were all first draw samples collected from outlets served by passively recirculated floor loops, and the three with the highest culture numbers were sampled from the same floor loop (F-floor). It is notable that the F-floor had the highest number of samples collected before and after the holiday breaks, implying it likely had more hot water demand than other floors due to this study. 

#### 3.2.3. No Culturable Legionella Detected during Controlled Stagnation

Controlled stagnation experiments were conducted because the full-scale system responded differently to the 3-week winter holiday and 7-week COVID lockdown periods of low water demand. Complete stagnation was imposed on two floors for 0 to 4 weeks. No culturable *L. pneumophila* was detected at any of the 26 outlets monitored before or after the imposed stagnation period. Total and intact cells increased after 1-week stagnation and remained elevated, and the percent of intact cells tended to decrease with time (Appendix A).

### 3.3. Pipe Surface Temperature Profiles

We installed temperature sensors on the surface of pipes in the main recirculation system, floor loops, risers, and at points of use, under existing insulation where applicable, as an indicator of recirculation and/or water demand to illustrate how water moved through the building. Note, water temperatures can be assumed to be 2–3 °C higher than pipe surface temperature due to thermal losses in heat transfer from the water to the pipe, insulation, environment, and sensor.

#### 3.3.1. Thermal Loss in the Main Recirculation System Indicates Growth Potential

While temperature set-points were generally reached in the hot water supply (median hot water supply pipe temperature was 42.8 and 56.4 °C on a typical 45 and 60 °C boiler operation day, respectively; Appendix A; Appendix A), significant temperature loss was observed within the main hot water recirculation loop. Median return pipe temperatures on the lowest floor, furthest away from the boiler in flow sequence, was 37.6 and 47.2 °C on 45 and 60 °C operation days (Appendix A). Over a typical 1-week period, the hot water supply temperature exceeded 50 °C approximately one third of the time, and the return line closest to the boiler never exceeded 50 °C (Appendix A). 

#### 3.3.2. Pipe Temperature Profiles on Floors Indicate Variable Growth Potential

Each floor loop had a characteristic passive recirculation profile. Median floor loop pipe inlet and outlet temperatures ranged from 34.8–37.4 °C and 42.4–48.1 °C on 45 and 60 °C boiler operations days except on Floor C, which had an erroneously closed ball valve in the middle of the floor loop preventing any passive recirculation (Appendix A). There was some convective mixing between passively recirculated floor loops and outlets they served located directly above the loop. The extent of this mixing was variable, with Floor C outlets (with no passive recirculation) remaining at ambient temperatures and pipes serving outlets on other floors stabilizing between room temperature and 32 °C depending on boiler operation (45 or 60 °C) and the specific outlet in question (Appendix A). Outlets served by risers remained at room temperature unless they were being used. 

### 3.4. Recommissioning Case Study and Experiments

#### 3.4.1. COVID-19 Recommissioning Flushing Temporarily Increased Legionella in the System

After flushing each outlet until steady hot temperatures were reached (or for 2 min, whichever occurred first), Eawag personnel returned to work in a limited capacity (<30% normal building occupancy). Again, contrary to expectations that *L. pneumophila* culture numbers would decrease after flushing, they remained constant in first draw samples (Figure 3A) and increased by 0.55–3.4 logs in flushed samples (Figure 3B) for two weeks after recommissioning flushing (Dunn Test with Bonferroni correction, *p*-value = 0.00017–0.022). Beginning in the third week of reoccupancy, flushed samples declined to <1000 MPN/L. In the sixth week of reoccupancy, median *L. pneumophila* in first draw and flushed samples returned to <100 MPN/L, similar to contamination levels observed during routine monitoring prior to the pandemic throughout 2019 (Kruskal-Wallis, *p*-values, 0.11–0.82). There were no differences between the outlets served by the floor loop relative to the risers in first draw (Appendix A, Kruskal-Wallis, *p*-values, 0.064–0.64) or flushed samples (0.068–0.81). *L. pneumophila* appeared to increase more rapidly in repeatedly sampled outlets after the buildings were reoccupied, but there was not enough data to statistically assess these trends.

#### 3.4.2. Thermal Barrier in Boiler Was Depleted during Recommissioning Flushing

During recommissioning flushing (approximately 08:00 to 16:00, 30 April 2020), the temperature of the water heater was depleted due to high water demand associated with flushing every outlet (Appendix A). Recommissioning flushing was paused to allow the boiler to recover several times, but thermally disinfecting temperatures were not maintained. The boiler outlet temperature was less than 60 °C for 70% of the recommissioning period and below 55 °C for 8%. The bottom of the boiler was always below 60 °C and was below 55 °C for 42% the of recommissioning flushing.

#### 3.4.3. Overall Bacteria Growth Responded to Changes in Water Demand 

Quantification of total (TCC) and intact (ICC) cell numbers after the COVID-19 lockdown and during recommissioning indicated regrowth and detachment of bacteria was occurring at unused outlets relative to in the main hot water system, particularly in outlets served by floor loops. While there were no differences in the numbers of cells at repeatedly compared to randomly sampled outlets (Appendix A), outlets fed by the floor loops tended to have ~0.5-log more TCC in first draw samples than outlets fed by risers (Figure 4A), which was eliminated in flushed samples representative of the hot water supply (Figure 4C). The proportion of cells with an intact membrane also tended to be higher in first draw samples of outlets served by the floor loops relative to the risers (Figure 4B). The proportion of intact cells was positively correlated with total cells in first draw samples (Spearman Rank, ρ = 0.42, *p*-value < 0.0001) but negatively correlated with total cells in flushed samples (ρ = −0.56, *p*-value < 0.0001). All of these observations indicate that conditions were suitable for bacterial growth and detachment during outlet stagnation and in passive recirculating loops, but not in the main hot water distribution system. 

#### 3.4.4. Intact Cells Trended Inversely with Legionella, but Not Significantly

The proportion of intact cells generally increased during the 5-week recommissioning monitoring in first draw (Figure 4B) and flushed samples (Figure 4D), as *L. pneumophila* generally decreased. However, *L. pneumophila* culture numbers did not have a significant relationship with total cells, intact cells, or the percent of intact cells in individual samples (Appendix A), possibly due to low sample sizes.

### 3.5. Boiler Flushing Experiments Highlight Potential for Unintended Consequences of Some Flushing Practices

Controlled flushing of 200% of the volume of water contained in the Eawag building boiler at two different flow rates and the two boiler set points repeated aspects of the surprising temporary increase in *L. pneumophila* observed during COVID-19 recommissioning flushing. 

While flushing at 5 Lpm, *L. pneumophila* gradually increased from below the Legiolert detection limit to 1970 MPN/L during flushing before dropping back below the detection limit after nearly 200% of the boiler volume was flushed. *Legionella* spp. and *L. pneumophila* gene copy numbers followed a similar pattern (Figure 5A). Hot water temperature measured at the outlet fluctuated between 39.7 and 44.2 °C during the 5 Lpm flushing, indicating that the boiler was able to heat fresh incoming water as it was being flushed, but it was not hot enough to thermally disinfect *Legionella*. 

While flushing at 95 Lpm, no culturable *L. pneumophila* was detected, presumably due to the high boiler temperature at the start of flushing. *Legionella* spp. and *L. pneumophila* gene copy numbers fluctuated during flushing by up to 2-logs, with spikes during the first 25% of the boiler volume and between 75–100% (Figure 5B). Hot water temperature was >55 °C until just before 100% of the boiler volume was reached, indicating the high flow rate through the boiler achieved near plug-flow behavior. Thus, thermally disinfecting temperatures were maintained during both spikes in gene copy numbers while flushing, but then rapidly decreased.

After flushing at both 5 and 95 Lpm, *L. pneumophila* increased by 1- to 2.5-logs in flushed samples in 2–3 weeks after boiler flushing occurred (Figure 5D). After flushing at 95 Lpm, first draw samples were also elevated by 1.5-logs.

## 4. Discussion

### 4.1. Water Demand Alone Is Not an Adequate Predictor of Legionella

The variable response of *L. pneumophila* to periods of low water demand observed in this study has been reported in other recent studies that assessed the impact of COVID-19 building closures. Liang et al. reported only a two-fold increase in the relative abundance of *Legionella* spp. via 16S rRNA amplicon sequencing after 2 months of low water demand, the same level of increase observed after just one overnight stagnation, suggesting that dramatic increases in *Legionella* abundance with extend periods of reduced demand did not occur [31]. Ye et al. reported the recovery of water quality parameters in university buildings after being unoccupied for 5 months. As the building returned to normal water demand, the occurrence rate and concentration of *L. pneumophila* did not decrease as expected if the closure resulted in growth [32]. De Giglio et al. reported that sample positivity rate did not consistently increase during 3-month closures, but levels of culturable *L. pneumophila* increased in all three wards monitored by a median of 175 to 5525 CFU/L [33]. Non-COVID-19 related case studies also demonstrate a varied impact of low water demand, with some instances where authors conclude that indicators of stagnation were not significantly associated with the decreased efficacy of chlorine dioxide or thermal disinfection [34,35] while others link stagnation to increased risk for legionellosis in a review of case-studies reports [36].

Stagnation as a catch-all term for water aging is problematic because it can occur at a single outlet or an entire system, for hours to years, and is coincident with many biotic and abiotic water quality changes that occur to different extents within or among different systems [8]. In this study, we specifically reserve the term “stagnation” to describe the lack of water demand and flow. As we define it, stagnation routinely occurs at end-use outlets when they are not being used, in some redundant systems (e.g., improperly sized parallel backflow preventors), and dead-legs (altered, abandoned, or capped plumbing that water cannot flow through). While stagnation can occur as a result of a building closure, it is more likely that some water demand and/or flow remains in the building due to occupancy of essential personnel, mechanical systems that use potable water supply (cooling tower make-up water), systems that automatically regenerate (some water softeners), or hot water recirculation systems. 

Risk of *Legionella* growth in building systems is also often over-simplified to be associated with stagnation. For instance, Galvada et al. has been cited as reporting low-use taps were associated with *Legionella* colonization [36]; however, the original study states that outlets that are not used daily *in poorly designed areas* (defined by low recirculating temperature) were 2× more likely to be positive and 3× more likely to have *Legionella* levels ≥ 1000 CFU/L than other outlets [37]. Völker et al. more assuredly demonstrated that low-use outlets and pipe length were associated with *Legionella* occurrence, but did not define or discuss thresholds for classifying outlets as stagnant [7], and other researchers have nuanced pipe distance, indicating that hot water pipe distances from the boiler alone was not a justifiable predictor variable and suggested that temperature profiles rather be used [38]. Thus, a better conceptual model of the impact of water demand on *Legionella* should incorporate other conditions present in the plumbing system.

### 4.2. Consideration of Simultaneous Growth Mechanisms Sheds Light on Variable Response

The growth of all microorganisms in drinking water is reliant on the presence of adequate nutrients and physiochemical conditions suitable for growth, which can vary significantly within a building. Thus, examining one growth mechanism at a time can only partially explain observations. For instance, examining fresh nutrient influx can partially explain *L. pneumophila* occurrence trends in this study. The low water demand and daily recirculation of hot water throughout the building system likely created nutrient-limiting conditions, resulting in low levels of *L. pneumophila* during regular demand periods in 2019 and the end of 2020. The low influx of nutrients may also be associated with the decrease in *L. pneumophila* levels during the COVID-19 lockdown. Starvation that reduces culturability of *Legionella* can occur within 5–11 days in pure culture [39,40], which can be considerably extended by the presence of host organisms [41,42]. With the recommissioning flushing, rapid influx of new nutrients may have stimulated new growth that then become nutrient limiting again over the next 2–3 weeks. 

However, normal operating conditions in this building and the recommissioning flushing did not only impact nutrient flux; importantly, they also controlled water temperature profiles. For instance, during normal occupancy, thermal loss and convective mixing along the primary and floor loop hot water recirculation systems consistently maintained suitable growth temperatures in the distal areas of the building. This may partially explain why first draw samples had elevated *L. pneumophila* levels relative to 5-min flushed samples, particularly on floor loops with passive recirculation that stabilized in the ideal growth range (32–42 °C). 

The recommissioning flushing resulted in a temporary thermal shock to the pipes flushed with 55–60 °C water, rapid boiler volume turnover, higher flow rates than pipes normally experience, and a possible shift in ecology. Temporary heat shock followed by periods of stagnation can support rapid growth of *L. pneumophila*, presumably due to dead-biomass created from thermal shock providing nutrients for amoeba (and thus *Legionella*) [4,43]. One study reported up to a 4-log increase of *Legionella* in biofilms, corresponding with a shift in the microbial community, after being fed heat-treated water [44] and regrowth of *Legionella* is frequently reported after thermal disinfection [4,5,43,45]. However, the stratified boiler was also depleted during flushing, resulting in the 60 °C thermal barrier being diminished and potentially dispersing *Legionella* throughout the system from the non-thermally disinfected biomass at the bottom of the tank [44,46,47]. In addition, the high flow rate associated with the simultaneous flushing of multiple outlets may have disturbed existing biofilms, also releasing and dispersing *Legionella*. This is consistent with the follow up study flushed at 95 Lpm, which demonstrated a ~2-log spike in *Legionella* spp. and *L. pneumophila* genes copies at the beginning of the flushing period. Finally, as building occupancy (and therefore water demand) increased during recommissioning, the proportion of intact cells increased as *L. pneumophila* generally decreased. This suggests that increased competition by other bacteria may have limited *Legionella* growth or was simply coincidental. Regardless of the precise cause, this study highlights that there are at least temporary unintended consequences to some flushing practices. 

Most perplexing in this study was the dramatic increase in *L. pneumophila* during the 2019 winter holiday break. While the conventional logic suggests the increase in stagnation associated with the break may have caused a rapid increase, water demand in this building was already likely very low and there were no such increases in *L. pneumophila* during winter holiday break 2020 or in the controlled stagnation experiments. Further, outlets that were likely more routinely used on F-floor consistently had higher levels of *L. pneumophila* (though statistical comparisons were not possible), directly contradicting the stagnation hypothesis. The sharp increase in *L. pneumophila* observed during winter break 2019 may rather be linked to uncontrolled and undocumented events related to construction associated with a new building adjacent to the study site. Construction activities, depressurization (maintenance, failures), and water treatment failures have been linked to elevated levels of *Legionella* occurrence and/or Legionnaires’ disease incidence [48,49,50,51]. While it is impossible to retrospectively link these events, such connections should not be ruled out and should be the subject of future research. 

### 4.3. Future Data Perspective

Without high resolution data to relate *Legionella* positivity and numbers to the history of use at individual outlets based on water demand patterns, researchers developed temperature profiling diagnostics to identify building plumbing sections with sub-optimal recirculation (<55 °C across the network), unintentional cross-connections (defective return valves causing hot and cold water mixing), and outlets with very long service connections (requiring high amounts of flushing to establish hot water) [5]. These profiles are very effective in identifying systemic issues with system performance, but need to be repeated to assess performance over time. Ideally, in-line temperature probes would be located throughout the plumbing system, but this approach requires alteration to the plumbing system. In this study, pipe surface temperature probes were able to characterize sub-optimal portions of the system. They could be more widely and easily applied than in-line sensors, and some practitioners may prefer this approach to more intrusive or time-intensive monitoring strategies. More strategically placed temperature probes may also be able to reflect water demand patterns, at least with respect to frequency of use and duration between uses. Those data would help to better define end-use stagnation in building plumbing and provide evidence for specific flushing recommendations. 

## 5. Conclusions

Reduced water demand associated with low building occupancy does not always cause *Legionella* growth, even when the building has been historically colonized by *Legionella*;Reduced water demand coincides with myriad other reactions in building plumbing that have to be accounted for when defining or describing building plumbing stagnation. In this study temperature profiles associated with hot water recirculation and convective mixing, nutrient depletion, water use patterns at individual outlets, and external disturbances were hypothesized to contribute to increased *Legionella* occurrence;Some flushing practices have the potential to temporarily increase *Legionella* occurrence. In this study rapid boiler turnover, high shear sloughing of biofilm associated with flushing many outlets simultaneously, and rapid nutrient influx were hypothesized to contribute to increased *Legionella* occurrence;Pipe surface temperature loggers can offer a non-invasive alternative to in-line probes while still providing continuous performance metrics.Future published work should carefully consider and define stagnation, and how it relates to other phenomenon occurring in the systems being studied.

## Figures and Tables

**Figure 1 microorganisms-10-00555-f001:**
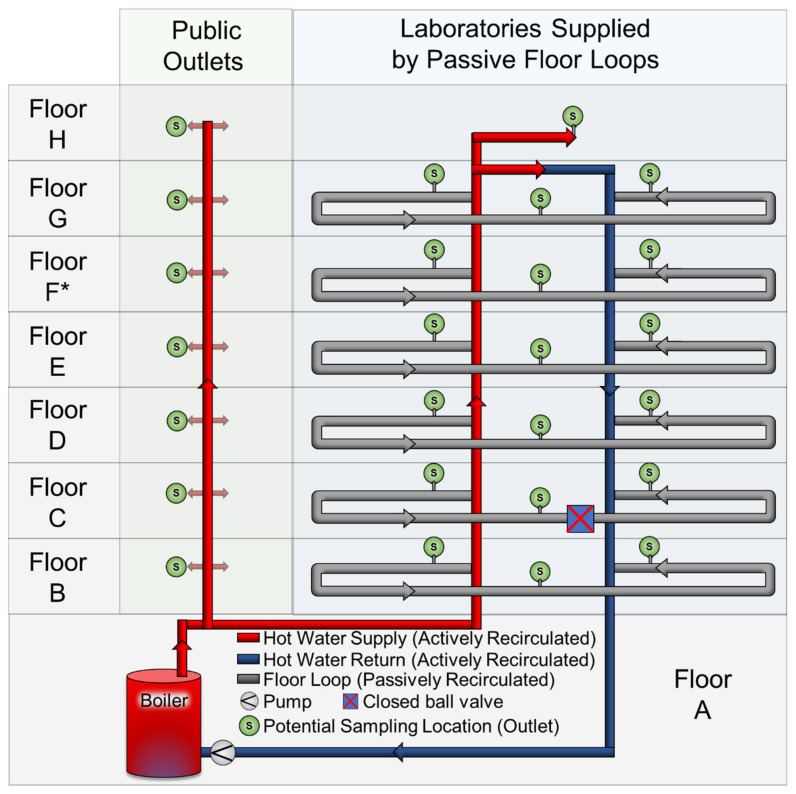
Simplified building plumbing schematic of the hot water system. The main boiler (1000 L) operates at 45 °C five days per week and at 60 °C on Tuesdays and Thursdays. Hot water is actively recirculated in the laboratory building (via the Hot Water Supply and Hot Water Return pipes) with a pump that operates from 05:00–19:00 7 days per week. In the laboratory building, hot water is passively recirculated on each floor; Floor C had an erroneously closed ball valve preventing circulation. In public areas in both the office and laboratory buildings (lavatories and kitchenettes), hot water is supplied directly from the boiler by eight risers (that each serve 1–16 outlets). * Floor F had higher water demand than other floors. See also Appendix A; Appendix A.

**Figure 2 microorganisms-10-00555-f002:**
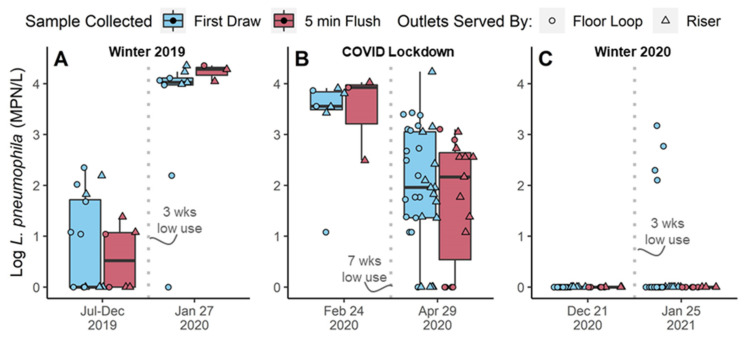
*L. pneumophila* culture numbers in hot water samples collected on a day the boiler was operated at 45° in first draw (blue) and 5-min flushed (red) samples before and after periods of low water demand due to (**A**) winter holiday break in 2019, (**B**) COVID Lockdown, and (**C**) winter holiday break in 2020. Boxplot bodies reflect the 25th, 50th, and 75th percentiles; whiskers represent 1.5 times the interquartile range. Raw data is overlaid onto the boxplots. ◦ indicates samples collected from passive floor loops (laboratory outlets); ∆ indicates samples collected from hot water risers (outlet in lavatories and kitchenettes).

**Figure 3 microorganisms-10-00555-f003:**
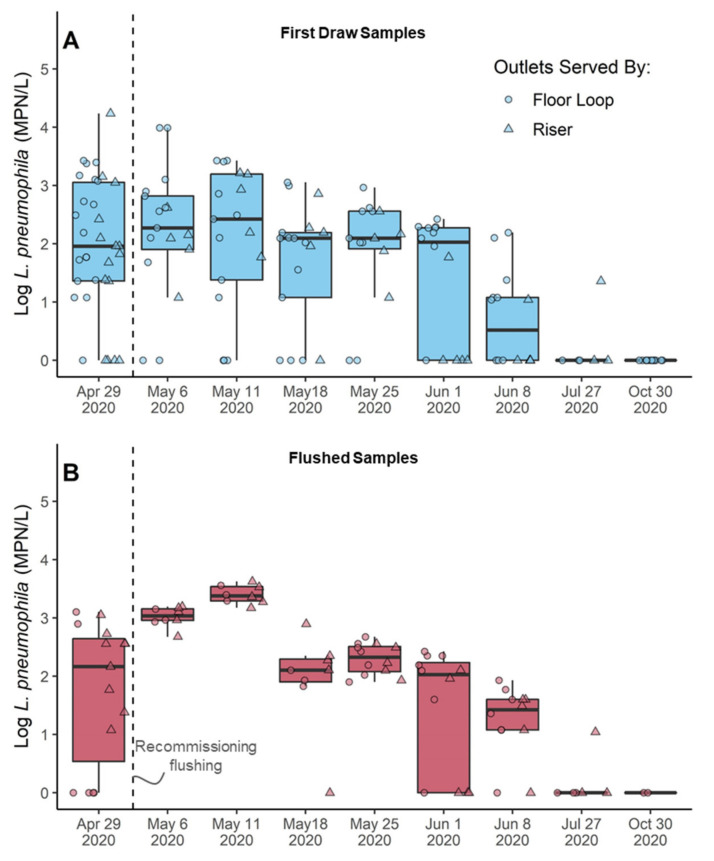
*L. pneumophila* culture numbers in hot water samples collected on a day the boiler was operated at 45 °C in (**A**) first draw and (**B**) 5-min flushed samples just before and several weeks after recommissioning flushing. Boxplot bodies reflect the 25th, 50th, and 75th percentile; whiskers represent 1.5 times the interquartile range. Raw data is overlaid onto the boxplots. ◦ indicates samples collected from passive floor loops (laboratory outlets); ∆ indicates samples collected from hot water risers (outlet in lavatories and kitchenettes).

**Figure 4 microorganisms-10-00555-f004:**
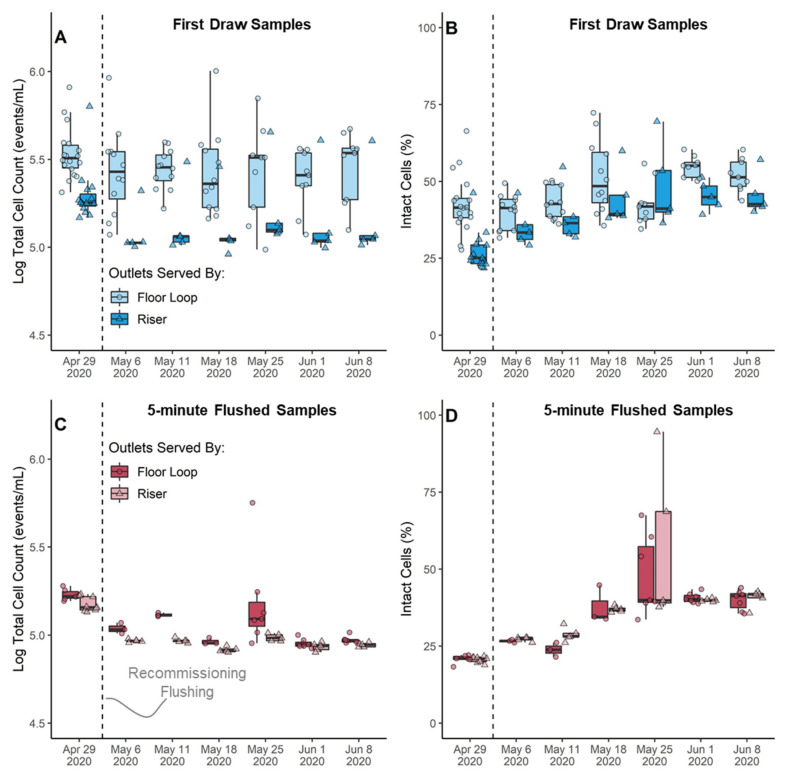
Total cell counts in (**A**) first draw and (**C**) 5-min flushed samples as measured by flow cytometry in events/mL. Percent of intact cells in (**B**) first draw and (**D**) 5-min flushed samples calculated by dividing intact cells by total cells in each sample. Boxplot bodies reflect the 25th, 50th, and 75th percentile; whiskers represent 1.5 times the interquartile range. Raw data is overlaid onto the boxplots. ◦ indicates samples collected from passive floor loops (laboratory outlets); ∆ indicates samples collected from hot water risers (outlets in lavatories and kitchenettes).

**Figure 5 microorganisms-10-00555-f005:**
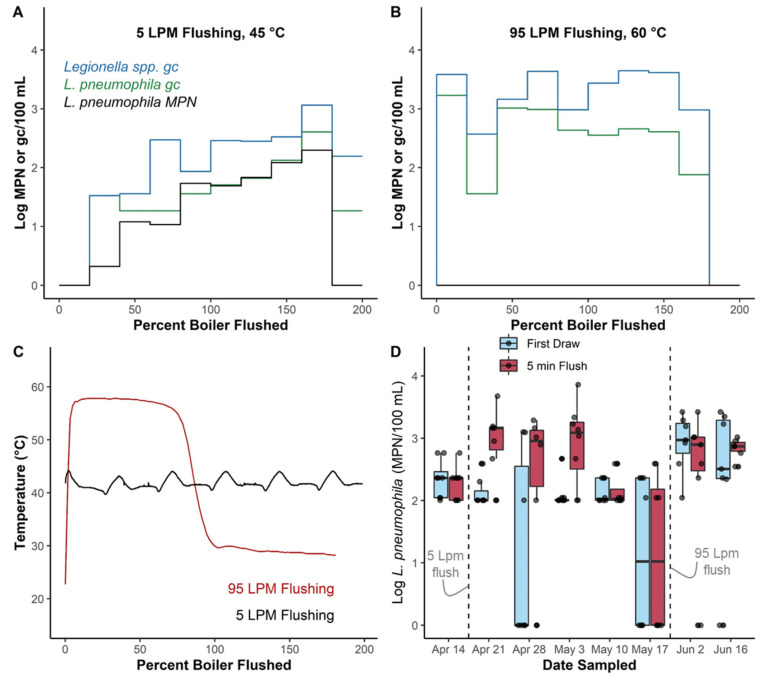
*L. pneumophila* culture and *Legionella* spp. and *L. pneumophila* gene copy numbers as a function of boiler volume flushed (**A**) at one outlet at 5 Lpm or (**B**) at 95 Lpm; (**C**) water temperature profiles during flushing; (**D**) *L. pneumophila* culture numbers in 8 first draw and 8 5-min flushed samples collected from the same outlets before and after flushing at 5 and 95 Lpm. Boxplot bodies reflect the 25th, 50th, and 75th percentile; whiskers represent 1.5 times the interquartile range. Raw data is overlaid onto the boxplots.

## Data Availability

Data is available in the Appendix A.

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
