# Peer review of "Variable Legionella Response to Building Occupancy Patterns and Precautionary Flushing"

_microorganisms, 2022, doi:10.3390/microorganisms10030555_

Round 1

Reviewer 1 Report

Comments for Authors

The manuscript is well written and well structured, and the topic is very interesting. I think it could be the starting point for updating guidelines and standards addressing Legionella risk management in buildings.

The Authors quantified legionellae with the Legiolert method that is specific only for Legionella pneumophila. Discussion and conclusions, therefore, must refer only to the pneumophila species and this limitation has to be stated by the Authors in the text. It cannot be excluded that stagnation may lead to an increase in species other than pneumophila or, conversely, the flushing may lead to a decrease in their counts. Authors should check overall the text and substitute “Legionella” with “L. pneumophila” where it is necessary.

In my opinion, the manuscript is suitable for publication after minor revisions.

Minor revisions:

Material and Methods section

Lines 195-197. IDEXX Legiolert method must be described in detail and at least one bibliographic reference added.

Lines 199. Specify on which and how many samples the molecular analysis was carried out.

Results

The Authors express the results almost always in terms of L. pneumophila concentration, but they should evaluate the contamination also in terms of number/percentage of positive points before and after the COVID lockdown and winter holiday breaks, especially that of December 2019, and after COVID recommissioning flushing.

Discussion

Line 460. The correct surname is De Giglio, not Giglio.

Line 503. Change the second “34” with 35!

Author Response

We thank the reviewer for their comments. 

Line 195-197: We have added a reference to the manufacturer details for the Legiolert kit. We also added a sentence about the quality of the kit and appropriate references to studies that have evaluated the specificity and sensitivity of the assay with respect to culture. Specifically, it tends to correlates well but trends slightly higher (1.2X) than culture and has a low (3-4%) false positive rate. 

Line 199: The ddPCR was performed only on the samples associated with full turnover of the boiler. These included 10 samples while flushing at 5 Lpm and 11 samples flushing at 95 Lpm (in addition to the positive and negative controls mentioned). This detail was added to the manuscript. 

Results: We have included positivity results throughout this section of the results. However, because our analysis focuses heavily on concentration and the drastic changes in positivity are somewhat self-evident in the study, we do not judge we need to elaborate on their interpretation. 

460 & 503: Both changes made; thank you!

Reviewer 2 Report

In the manuscript "Variable Legionella Response to Building Occupancy Patterns and Precautionary Flushing" authors present field-scale research. Conducting such studies is a challenge and the authors have planned everything in detail and presented it in the obtained results. Water flow and stagnation are important factors in the colonization of Legionella and the authors have presented this. The  material of water pipes that was placed in the building and the presence of other OPPPs and amoebas remain questionable to me, so this could also be commented on in the discussion.

Author Response

We thank the reviewer for their comments. We are not able to obtain detailed information about the installation materials behind the walls (mostly, we believe it is stainless steel). We do document in the methods the materials that were installed from the stop cock to the faucets. However, because the materials did not change from before, during, or after the stagnation or flushing interventions, we do not judge that this would alter or influence our interpretation of the data. Thus, we have not made additional references to the building materials in the manuscript.